# Identification of Estradiol Benzoate as an Inhibitor of HBx Using Inducible Stably Transfected HepG2 Cells Expressing HiBiT Tagged HBx

**DOI:** 10.3390/molecules27155000

**Published:** 2022-08-06

**Authors:** Jingjing He, Jingwen Wu, Jingwen Chen, Shenyan Zhang, Yifei Guo, Xueyun Zhang, Jiajia Han, Yao Zhang, Yue Guo, Yanxue Lin, Weien Yu, Yide Kong, Zhongliang Shen, Richeng Mao, Jiming Zhang

**Affiliations:** Shanghai Key Laboratory of Infectious Diseases and Biosafety Emergency Response, Department of Infectious Diseases, Shanghai Institute of Infectious Diseases and Biosecurity, National Medical Center for Infectious Diseases, Huashan Hospital, Fudan University, Shanghai 200040, China

**Keywords:** estradiol benzoate, HBx, hepatitis B virus

## Abstract

HBx plays a significant role in the cccDNA epigenetic modification regulating the hepatitis B virus (HBV) life cycle and in hepatocyte proliferation and carcinogenesis. By using the sleeping-beauty transposon system, we constructed a tetracycline-induced HBx-expressing stable cell line, SBHX21. HBx with a HiBiT tag can be quickly detected utilizing a NanoLuc-based HiBiT detection system. By screening a drug library using SBHX21 cells, we identified estradiol benzoate as a novel anti-HBx agent. Estradiol benzoate also markedly reduced the production of HBeAg, HBsAg, HBV pgRNA, and HBV DNA in a dose-dependent manner, suggesting that estradiol benzoate could be an anti-HBV agent. Docking model results revealed that estradiol benzoate binds to HBx at TRP87 and TRP107. Collectively, our results suggest that estradiol benzoate inhibits the HBx protein and HBV transcription and replication, which may serve as a novel anti-HBV molecular compound for investigating new treatment strategies for HBV infection.

## 1. Introduction

The Hepatitis B virus (HBV) is one of the leading causes of liver fibrosis and hepatocellular carcinoma (HCC) and places a heavy burden on healthcare costs worldwide. According to World Health Organization estimates, 296 million patients have chronic hepatitis B (CHB) worldwide, with 1.5 million new infections occurring annually [1]. The availability of potent antiviral therapies, consisting of nucleos(t)ide analogues and interferon treatment, leads to the long-term inhibition of HBV replication [2]. Nonetheless, both treatments have certain drawbacks [3]. Studies have shown that after long-term nucleos(t)ide treatment, silent HBV covalently closed circular DNA (cccDNA) remains latent in the nucleus, which makes eradication of the virus difficult and leaves the possibility of reactivation. Pegylated interferon can achieve HBsAg loss in a certain group of patients, but the treatment requires inconvenient subcutaneous injections and may lead to significant side effects in some patients [2]; therefore, new treatment strategies need to be explored.

The cccDNA minichromosome is a template for the transcription of viral mRNAs and the 3.5 kb pre-genomic RNA (pgRNA), which is the reverse-transcribed viral DNA. The cccDNA transcription activity is regulated by the epigenetic modifications of DNA and histones that bind to cccDNA [4,5]. The HBV regulatory HBx protein is translated from the 0.7 kb mRNA and plays a crucial role in cccDNA activity [6]. Previous studies have shown that HBx binds to cellular-damaged DNA-binding protein 1 (DDB1) and regulates the degradation of Smc5/6 to promote cccDNA activity. DDB1 is an adaptor protein for the cullin 4A (CUL4)–Really Interesting New Gene (RING)–E3 ubiquitin ligase (CRL4) complex, while HBx could mimic cellular DDB1-cullin-4-associated factor (DCAF) receptor proteins to bind with DDB1 [7]. HBx also inhibits the recruitment of other host inhibitory factors, including PRMT5, APOBEC3B, and LINC01431 [8,9,10]. Moreover, HBx also regulates the cccDNA transcriptional activity by preventing transcriptional repressor recruitment on the cccDNA, including SIRT1 and HDAC1 [11,12]. HBx not only contributes to persistent HBV infection but also modulates several cell-signaling pathways, such as P53, mitogen-activated protein kinase (MAPK), nuclear factor kappa-light-chain-enhancer of activated B cells (NF-kB), and the Janus kinase signal transducer and activator of transcription (JAK-STAT) pathways, thereby promoting cell proliferation and inducing carcinoma formation [13,14,15,16]. Anti-HBx could be a potential antivirus strategy for HBV infection, but because HBx is an intracellular protein and most antibodies against HBx are not sensitive enough, the discovery of inhibitors of HBx would be facilitated by the ability to screen large chemical libraries in a short time for molecular compounds against HBx. To achieve batch screening of HBx-inhibiting drugs, we attempted to establish a cell line that produces a high level of HBx that can be easily detected. We utilized the quantitative NanoLuc-based HiBiT detection system. A luciferase complementation was split into two NanoLuc fragments, a 1.3-kDa peptide (11 amino acids) named HiBiT and an 18-kDa polypeptide named Large BiT (LgBiT). The HiBiT peptide was attached to the N-terminal of HBx and could be measured by chemical fluorescence by binding to the LgBiT. The HiBiT tag is short and does not affect the HBx function; therefore, with the help of a sleeping-beauty transposon system, such a cell line establishment was deemed to be safe and fast.

We constructed and utilized an HBx-expressing SBHX21 cell line for drug library screening. According to our results, the sex hormone molecular estradiol benzoate could reduce intracellular HBx levels. Estradiol benzoate suppressed HBV replication and transcription in the cellular model of the HBV replication using HepG2.2.15 cells and HepG2 transfected cells. Protein molecule interaction modeling indicated that estradiol benzoate forms covalent bonds with HBx at TRP87 and TRP107. This sequence covers the H-box of the HBx sequence from Ile88 to Leu100, which docks with DDB-1. These observations provide a novel perspective to understand the role of estradiol benzoate on HBx in HBV infection.

## 2. Results

### 2.1. Establishment of the SNHX21 Cell Line

The 544bp HBx sequence from pGFP-HBx was generated into a pBIT3.1 N CMV/HiBiT/Blast Vector using ScaI and Hind restriction sites. The pSBtetHX plasmid was constructed by exchanging the luciferase sequence of pSBtet-Hyg with the HiBiT-HBx sequence using two slightly different SfiI restriction sites (Figure 1A). The pSBtet-Hyg plasmid had two inverted terminal repeats (ITRs) sequences, which resulted in the integration of the genes flanked by ITRs with the assistance of SB100X transposases expressed by pCMV(CAT)T7-SB100. The integrated genes included the controllable TCE (tetracycline response element and minimal CMV-promoter enhanced) promoter, which enhanced the expression of HiBiT-HBx in a tetracycline-inducible manner, whereas the rTetR sequence expressed the rtTA protein and the hygromycin gene. A pSBtetHX plasmid was shown to express HBx in a Tet-inducible manner (Figure 1C).

Then, a stable HiBiT-HBx–expressing HepG2 cell line was generated. Resistant clones were selected in the presence of hygromycin at a concentration of 300 µg/mL. After dilution and culturing for two weeks without the selection medium, cell clones that were visible to the naked eye were isolated and expanded (Figure 1D). Most clones were able to support high replication of HiBiT-HBx, as confirmed by fluorescence detection. After a continued passage, the cell clone termed SBHX21 was expanded and examined for the stable production of HiBiT-HBx. Expression of HiBiT-HBx was induced by doxycycline (Dox) administration for 48 h. The cell lysates were separated and analyzed with a HBx antibody (Figure 1E). An induced protein expression was also measured by relative luminescence units (RLU) using a HiBiT Lytic Detection System (Figure 1F); therefore, the inducible expression of intracellular HBx could be measured by luminescence generated by the combination of HiBiT and the LgBiT Protein.

### 2.2. Drug Screening Results Identify Estradiol Benzoate as an Inhibitor of HBx

Thus far, the SNHX21 cell line has been adopted to quickly measure the level of intracellular HBx protein, which has made it convenient to be used for the screening of anti-HBx molecular compounds, siRNA, or intracellular genes. Our study screened the drug library consisting of a collection of 1403 FDA-approved drugs. The drug-screening process had three periods. First was the Dox induced HiBiT-HBx protein expression for two days. Second, the media were changed, and the drugs were added at a final concentration of 10 µM without Dox. Third, the cells were lysed using the HiBiT Lytic Detection System after 12 h. Ten minutes after the lysis, the level of HBx was measured by the level of luminescence (Figure 1B). Among nearly 1000 compounds, estradiol benzoate was confirmed to inhibit HBx in a dose-dependent manner by luminescence measurement (Figure 1G). Estradiol benzoate is a synthetic estrogen medication considered to be a natural and bioidentical form of estrogen. We further examined the inhibitory function of the major estrogen steroid hormone in females, 17β-estradiol, which also exhibited a dose-dependent reduction in intracellular HBx as shown by the luminescence measurement (Figure 1H). The CCK8 experiment results indicated that the working concentrations of drugs were much lower than the drug cytotoxic levels (Figure 1I,J). In the HiBiT lytic detection experiment, we found that 35 µM and 55 µM estradiol benzoate significantly inhibited HBx. Next, we validated this conclusion by a Western blot assay (Figure 1K). The ImageJ analysis revealed that the relative expression of HBx in the 35 µM concentration group was about half of that in the DMSO group (Figure 1L). The results demonstrated that estradiol benzoate had an inhibition function of HBx.

### 2.3. Estradiol Benzoate Inhibits HBV Activity

HBx plays a momentous role in the HBV life cycle, which is essential for virus replication and cccDNA transactivation. We detected the inhibitory function of estradiol benzoate and 17β-estradiol on HBV protein translation, HBV RNA transcription, and HBV DNA replication in the HepG2.2.215 system and 1.3mer HBV transfection system (Figure 2A). The ELISA results showed that those two drugs inhibited the secretion of exocellular HBeAg and HBsAg in a dose-dependent manner (Figure 2B,C), while supernatant HBV DNA did not show a significant decline (Figure 2D). In the 1.3mer HBV X-null plasmid transfected cells, supernatant HBsAg, HBeAg, and HBV DNA did not significantly differ between the groups (Figure 2). We further evaluated the inhibitory effect of estradiol benzoate and 17β-estradiol on HBV RNA transcription. After 72 h of treatment, the level of pgRNA (3.5 kb) was analyzed by RT-qPCR of the relative gene expression level to β-actin. Compared with the DMSO group, the levels of pgRNA of the drugs-treated group in HepG2.2.15 and transfected HepG2 cells were inhibited in a dose-dependent manner, and when the concentration reached 55 µM, the pgRNA demonstrated a significant reduction (Figure 3A). In the 1.3mer HBV X-null plasmid transfected cells, pgRNA did not significantly differ between groups (Figure 3C). To further verify this conclusion, a Northern blot was utilized to explore the effect of estradiol benzoate on HBV RNA in the HepG2.2.15 and 1.3mer HBV–transfected HepG2 cells. The Northern blot results showed that the HBV RNA synthesis was influenced by estradiol benzoate, as 3.5 kb RNA, 2.4/2.1 kb RNAs, and 0.7 kb RNA declined in a dose-dependent manner (Figure 3D). To evaluate the effect of estradiol benzoate on the cccDNA, HepG2.2.15 and transfected HepG2 cells treated with estradiol benzoate were examined by qPCR following a Hirt extraction. The results showed that the cccDNA demonstrated a dose-dependent reduction (Figure 3B). The Southern blot results also showed that estradiol benzoate inhibited various forms of HBV intracellular DNA replication intermediates in 1.3mer HBV plasmid–transfected HepG2 cells treated with estradiol benzoate for three days, compared with the DMSO group (Figure 3E). Taken together, these results demonstrated that estradiol benzoate had a downregulation effect on HBV translation, transcription, and replication.

### 2.4. Estradiol Benzoate and HBx-Binding Model

Estradiol Benzoate is the synthetic benzoate ester of estradiol (Figure 4A). HBx protein is a soluble protein with a large proportion of hydrophobic structures, which will form polymers with each other in solution or combine with other protein structures (Figure 4C). We assumed that estradiol benzoate represses HBV replication and translation through the inhibition of the HBx protein. The HBx structure has been modeled by I-TASSER (job id: S678536). Model 1 of the HBx (C-score = −4.40, estimated template-modeling (TM)-score = 0.25 ± 0.08, and estimated root mean square deviation (RMSD) = 15.4 ± 3.4Å) was adopted to predict the interaction of HBx and estradiol benzoate (Figure 4B). The protein was processed with the Autodock4.2 software. The docking for the HBx and estradiol benzoate was calculated using the Autogrid and Autodock programs, and the results were visualized using Pymol. The docking results showed that TRP87 and TRP107 of the HBx protein could form hydrogen–oxygen bonds with estradiol benzoate (Figure 4D). Interestingly, HBx binds with DDB1 at its larger pocket enclosed by its BPA-BPC double propeller fold, using a three-turn α-helix (termed H-box), which is essential for its reported stimulatory activities in cultured cells and might be required for efficient viral infection [7] (Figure 4E). This α-helical range from Ile88 to Leu100 closely overlapped the sequence from TRP87 to TRP107. Intermolecular docking between the HBx and estradiol benzoate may block the H-box from reaching out to bind with DDB-1. DDB-1 can bind to other adaptor proteins and perform its E3 protein function. The RNA-seq results revealed that CUL4-DDB1-related downstream genes were reduced after being treated with estradiol benzoate. The gene expression of EXO1, OSR1, AFP, CRB2, CDC45, ESCO2, MCM10, GPC3, CENPI, MYB, ATF3, and PER2 from the HepG2.2.15 cells all showed a decrease after being treated with 55 µM of estradiol benzoate for two days, indicating the activation of other DDB1-CUL4-related signals.

## 3. Discussion

HBx binds to the nuclear cccDNA minichromosome, initiating and maintaining HBV transcription and driving replication. The screening of drugs against HBx could be used as a promising treatment strategy for eliminating HBV infection and reducing the risk of HCC. Previous studies have reported that NQO1 could regulate the proteasomal degradation of HBx, thereby inhibiting HBV cccDNA activity [17]. Other studies have reported that intracellular restriction genes, including TRIM21, ISG15, and TRIM31, could eliminate HBx [18,19,20]. Another inhibition strategy against HBx could include short hairpin RNAs against HBx [21]. The results from those studies showed that therapeutic strategies against HBx are promising. We constructed the SNHX21 cell line, which was able to express HBx in a Tet-induced manner and making it possible to quickly detect the HBx levels using the Nano Glo HiBiT lytic detection system. This cell line may allow for fast testing of a large number of molecule compounds in industrial situations against HBx. SBHX21 cells could also be a benefit for the apprehension of HBx function and its protein–protein and protein–molecular interaction, which delineate a panorama of the cccDNA activity regulatory system.

After drugs screening, we identified estradiol benzoate as an HBx inhibitor. Both the HiBiT lytic detection experiment and Western blot results showed that estradiol benzoate downregulated the HBx levels. The ELISA, Northern blot, and Southern blot experiments also demonstrated that estradiol benzoate suppressed cccDNA transcription and replication activity. The observation time may not have been long enough to detect changes in the HBV capsid assembly; therefore, supernatant HBV DNA did not show a significant reduction in the early transfection period. The cccDNA was from pgRNA reverse-transcription. The ability of HBx to regulate the cccDNA activity was limited until cccDNA was formed in the nucleus. We assumed that a large amount of pgRNA and core particles were already formed in the first 24 h after transfection, and that a viral particle was secreted out of the cells afterward. The observation time of the extracellular HBV DNA theoretically can be put back to the observation time of the intracellular DNA. Taken together, estradiol benzoate could be a novel small molecule compound for use against HBx and HBV. For example, the inhibitory function of estradiol benzoate may occur by interacting with HBx. HBx contains 154 residues, most of which are hydrophobic. The N-terminal fragment of HBx remains insoluble, and the C-terminal portion adopts several α-helices. One α-helix domain named H-box motif (Ile88 to Leu100), could dock to the top surface of the DDB-1 BPC domain and regulate the CUL4-DDB-1 E3 complex productiveness [7]. Docking using AutoDock 4 illustrated that estradiol benzoate binds with HBx at TRP87 and TRP107, which overlap the H-box of HBx. Those covalent bonds may prevent HBx from forming polymers and the H-box from binding to DDB-1, and as a consequence, intervene reprograms of the CUL4 E3 ligase; however, this assumption needs further experimental evidence. In this study we mainly illustrated a phenomenon that estradiol benzoate inhibits HBx and HBV activity. Direction for further experiments that prove estradiol benzoate inhibits a HBx-DDB-1 interaction directly and regulates host factors such as Smc5/6, include an immunoprecipitation-Western blot analysis, in vitro Glutathione S-transferase (GST) pull-down assay, and a degradation experiment of Smc5/6 after drug treatment. Other possible mechanisms that explain why estradiol benzoate and 17β-estradiol decrease HBx and inhibit HBV viral activity are also needed to be investigated. Estradiol may mediate its anti-HBV function through the Estrogen receptor-α (ER-α) signaling pathway as it does in HIV replication [22], or through activation of the E3 ubiquitination pathway by estradiol benzoate which may contribute to a decrease in HBV-related protein [23]. A distribution and a significant decrease in NTCP were observed in EB-induced cholestatic rats, as previously reported [24], which is the primary receptor of the HBV entrance of cells. Whether there are other possibilities for estradiol benzoate regulating HBV infection are open for further investigation.

Sex hormones have been reported to be associated with HBV infection and the risk of HCC. The prevalence of HBV infection is significantly higher in men than in women, while the risk of developing HCC is two to four times higher in male than in female patients [25]. An analysis of estrogen receptors-α (ER-α) genomic region of single-nucleotide polymorphisms (SNPs) in a Chinese population showed that the ESR1 29T/T genotype had an increased susceptibility to persistent HBV infection [26]. An analysis of polymorphisms also indicated that individuals with the PvuII T/C genotype of ER-α might be greater virological responders to entecavir than those carrying the T/T and C/C genotypes [27]. Estrogen treatment had been reported to suppress HBV DNA, RNA, and HBeAg expression in male athymic mice bearing 2.2.15 cells [28]. Previous studies also reported a correlation between ER-α and HBx. It has been shown that HBx can bind to estradiol receptor-α (ER-α), inhibiting the activity of LINC01352 [29]. HBx, ER-α, and HDAC1 could form a ternary complex, thereby reducing the transcriptional activity of ER-α [30]. The interaction between ER-α and hepatocyte nuclear factor (HNF)-4α prevents HNF-4α from binding to the Enhance I regain of the HBV gene; therefore, the upregulation of ER-α reduces HBV transcription [31]. The inhibitory function of estrogen on virus infection has also been reported. Estradiol inhibits the Hepatitis C virus (HCV) transmissibility and HCV RNA replication through estradiol interaction with ER-α [32,33]. Estradiol was also proven to manipulate the immune system and intracellular intrinsic immune signaling. In the meantime, clinical usage of estradiol benzoate may be limited since estradiol benzoate, and 17β-estradiol are hormonal drugs that have a wide-range of effects. In HBV infection, our findings may serve as an exploration of the anti-HBV molecular mechanism. Then, the conditions for clinical use can be explored, which still has a long way to go. Further studies are needed to demonstrate whether there are other mechanisms by which estradiol benzoate inhibits HBx.

## 4. Materials and Methods

### 4.1. Cell Culture and Transfection

The HepG2 cell line was obtained from ATCC (HB-8065). The human hepatoma Huh7 cell line and HepG2-derived 2.2.15 cell line (constitutively producing HBV Dane particles) [34] were kindly provided by the Cell Bank, Chinese Academy of Sciences (Shanghai, China). All cells were cultured in Dulbecco’s modified Eagle’s medium (DMEM), supplemented with a final concentration of 10% fetal bovine serum (FBS), penicillin and streptomycin (1000 µg/mL final) at 37 °C in 5% carbon dioxide. The SBHX21 cell lines were maintained in hygromycin (150 µg/mL) containing supplemented DMEM. The plates were coated with 50 µg/µL Collagen Type I (Corning 354236) before culturing the HepG2 cells, HepG2.2.15cells, and SBHX21 cells. The HepG2 cells and Huh7 cells were transfected with plasmids using a Lipofectamine 3000 Transfection Reagent (Invitrogen, Carlsbad, CA, USA) following the manufacturer’s protocol.

### 4.2. Plasmids

The pBIT3.1 N CMV/HiBiT/Blast Vector (N2361) was purchased from the Promega Company. The pGFP-HBx was a gift from Prof. Xin Wang (Addgene 24931) [35]. The pSBtet-GH was a gift from Prof. Eric Kowarz (Addgene 60498) [36]. The p pCMV(CAT)T7-SB100 was a gift from Prof. Zsuzsanna Izsvak (Addgene 34879) [37]. The 1.3mer HBV plasmid was constructed and maintained in our laboratory, which had an HBV sequence (gene type B, NCBI Nucleotide: Hepatitis B virus strain 536207) with its self-promoter cloned to pcDNA3.1 zero (-). The 1.3mer HBV X-null replication plasmid was a gift from Prof. Zahid Hussaina (Addgene 65461) [38]. The HiBiT-X6 plasmid was generated by cloning the HBx sequence from the pGFP-HBx into the pBIT3.1 N CMV/HiBiT/Blast Vector. The primer sequences were as follows: forward 5′-ATATCGAGCTCCATGGCTGCTCGGGTGT-3′, reverse 5′-CCCAAGCTTGGGTTAGGCAGAGGTGAAAAA-3′. The pSBTetHX plasmid was designed by replacing the luciferase fragment of pSBtet-GH with the HiBiT-HBx fragment cloned from the HiBit-X6 plasmid. The prime sequences used for copying the HiBiT-HBx fragment were as follows: forward 5′-AGGCCTCTAGGCCGCCACCATGGTGAGCGGCTG-3′, reverse 5′-AGGCCTGACAGGCCTTAGGCAGA GGTGAAAAAG′-3′. The plasmids’ sequences were verified and are available upon request.

### 4.3. Cell Line Establishment

The HepG2 cells were cultured on 6-well plates and were a co-transfection of 1.9 µg pSBtetHX vector and 100 ng SB100X transposase vector per well using Lipofectamine 3000. Twenty-four hours after transfection, the cells from one well were digested and cultured in a 10 cm dish, subjected to 300 µg/mL hygromycin, and selected for two weeks. The cell masses were digested, counted, diluted, and transferred to a new 10 cm dish until all cells were separate from each other. The cells were cultured for two more weeks without selection. Then, individual clones were identified, and they were isolated using cloning cylinders (Merk C1059) and maintained in DMEM. After a functional verification, the SBHX 21 cell line that could express HiBiT-HBx under induction was selected for subsequent research.

### 4.4. Drug Library Screening

The drug library consisted of a collection of 1403 Food and Drug Administration (FDA)-approved drugs and was purchased from the TOP SCIENCE company, Shanghai, China. The drugs were dissolved in dimethyl sulfoxide (DMSO) with a concentration of 10 mM and were stored at −30 °C. The drug dilutions were prepared from a stock solution with DMEM. The SBHX21 cells were cultured in 96-well Flat Clear Bottom White Polystyrene Microplates (Corning) at 1 × 104 cells per well in DMEM containing Dox (1 µg/mL) for two days. Then, diluted drugs and fresh DMEM (without Dox) were added to each well to a final concentration of 10 µM of each drug. After 12 h, the cells were lysed using the Nano-Glo^®^ HiBiT Lytic Detection System (Promega, Madison, WI, USA) following the manufacturer’s protocol. Briefly, 1 µL of LgBiT Protein and 2 µL of HiBiT Lytic substrate were diluted into a 100 µL HiBiT Lytic Buffer at room temperature, with mixing by inversion to make the HiBiT Lytic Reagent. Cells from each well were washed with PBS twice, added with 100 µL of PBS and 100 µL of Lytic Reagent, and mixed on an orbital shaker (600 rpm) for 5 min. Ten minutes later, measurements of the luminescence using a SpectraMax Paradigm were conducted.

### 4.5. Cytotoxicity Assay

The cytotoxic effects of estradiol benzoate (MCE Company HY-B1192, Shanghai, China) and 17β-estradiol (MCE Company HY-B0141, Shanghai, China) were assessed by the Cell Counting kit-8 (CCK8) assay (MCE Company HY-K0301, Shanghai, China). The HepG2 cells were cultured in 96-well plates at 1 × 104 cells per well with three replicates per group in a concentration gradient of 1 µM, 50 µM, 100 µM, 150 µM, 200 µM, 250 µM, 300 µM, 350 µM, 400 µM, 450 µM, 500 µM, 550 µM, 600 µM, 650 µM, and 700 µM. The DMSO was normalized to 1% in all treatment groups. After 24 h, the medium was replaced with 10% CCK-8 diluted in fresh DMEM. CCK-8 solution and culture media (no cells) were added to the blank control. The absorbance (OD value) was recorded. Cell survival rate curves and IC50s were calculated using GraphPad Prism.

### 4.6. Western Blot

To analyze the expression of HiBiT-HBx, the SBHX21 cells were induced with Dox (1 µg/mL) for 48 h in a 12-well plate. Then, the cells were treated with different concentrations of estradiol benzoate and 17β-estradiol for 24 h without Dox. We used 0.5% DMSO DMEM as the control group. After treatment, the cells were lysed with 100 µL of RIPA buffer (Beyotime, Beijing, China). The cell extracts were separated on sodium dodecyl-sulfate polyacrylamide gel electrophoresis (SDS-PAGE) and blotted onto a nitrocellulose (NC) blotting membrane. The protein expression was analyzed using mouse monoclonal to Hepatitis B Virus X antigen (Abcam Eugene, Eugene, OR, USA) and was visualized with an Odyssey CLX System.

### 4.7. Quantitative Assay of Supernatant HBsAg, HBeAg, and HBV DNA

HepG2.2.15 cells were plated in 12-well plates. Then, estradiol benzoate and 17β-estradiol were added at final concentrations of 35 µM, 55 µM, and 75 µM for two days, while 1% DMSO DMEM was used as the control group. Each group had three replicates. The HepG2 cells were plated in 12-well plates and transfected with 1.3mer HBV plasmid. Twenty-four hours, the medium was replaced with drug-diluted DMEM at final concentrations of 35 µM, 55 µM, and 75 µM for two days. The culture media were harvested, centrifuged, diluted, and measured for HBeAg and HBsAg using an enzyme-linked immunosorbent assay (ELISA; Kehua Bio-Engineering, Shanghai, China). HBV DNA was analyzed using a Diagnostic Kit for Hepatitis B virus DNA (PCR-Fluorescence Probing; Sansure Biotech) following the manufacturer’s instructions. Briefly, the supernatant was centrifuged and treated with a sample release reagent (Sansure Biotech, Changsha, China). An amount of 5 µL sample release reagent was added to 5 µL supernatant. Then, 38 µL of the reaction reagent, 2 µL of the enzyme mixture, and 0.2 µL of the internal standard were added to 10 µL samples. The quantitative standard curve was established by setting the concentration of the standard sample at 4 × 107, 4 × 106, 4 × 105, and 4 × 104 IU/mL.

### 4.8. Viral RNA Analysis

HepG2.2.15 and HepG2 cells were cultured as described previously. After being treated with drugs for three days, the total cellular RNA from 12-well plates was lysed using TRIzol reagent (Invitrogen) following the instruction manual. The cDNA was generated using 5× FastKing-RT SuperMix (Tiangen Biotech, Beijing, China). The cDNA was subjected to real-time qPCR assay using FastFire qPCR PreMix (SYBR Green; Tiangen Biotech). The data were analyzed by the comparative CT (ΔΔCT) method, quantitated relative to the β-actin gene, and normalized to the control. The primer sequences were as follows: β-actin, forward 5′-GCACTCTTCCAGCCTTCC-3′, reverse 5′-GGTCTTTGCGGATGTCC-3′; pgRNA, forward 5′-GCCTTAGAGTCTCCTGAGCA-3′, and reverse 5′-GAGGGAGTTCTTCTTCTAGG-3′. The Northern blot procedures followed the DIG Northern Starter Kit. An RNA probe was generated from HBV plasmid pSP65+ that was constructed and maintained in our laboratory. The extracted RNA was separated by denaturing 1.5% formaldehyde agarose gel electrophoresis and transferred to a Hybong-Nþ membrane (Amersham Biosciences, Amersham, UK) in 20 SSC (Invitrogen, Carlsbad, CA, USA). The membrane was hybridized and detected using a DIG Wash and Block Buffer Kit. The signals were recorded using a ChemiDoc XRS+ system and Image Lab software.

### 4.9. Analysis of Viral DNA

HepG2.2.15 and HepG2 cells were cultured and treated as described previously. Three days later, the cccDNA-containing extrachromosomal nuclear DNA from a 12-well plate was prepared using the Hirt method [39]. The extracted DNA samples were resuspended in 20 µL ddH2O. Next, 2 µL of each sample was diluted with ddH2O to a final volume of 10 µL and used for RT-PCR amplification using the Diagnostic Kit for Hepatitis B virus DNA (Sansure Biotech). Samples values were normalized to the control group. A Southern blot of the HBV DNA followed using a Roche DIG High Prime DNA Labeling and Detection Starter Kit II. Briefly, the cells from transfected HepG2 cells on a 6-well plate were lysed in 400 µL lysis buffer (10 mM HEPES, pH 7.5, 100 mM NaCl, 1 mM EDTA, and 1% NP-40), precipitated with 125 µL of 26% PEG solution, followed by 10 U DNase I digestion for 2 h, a 200 µL of Proteinase K Buffer (35 mM Tris-HCl, pH 7.9, 35 mM EDTA, 175 mM NaCl, 1.5% SDS, and 1.35 mg/mL Proteinase K) digestion for 6 h, phenol extraction, and then ethanol precipitation. The purified DNA was dissolved in a TE buffer and separated in 1.2% agarose gel. The gel was denaturalized, neutralized, and transferred to a Hybong-Nþ membrane in 20 SSC. The membrane was cross-linked by UV and incubated in a full-length HBV genome probe-containing DIG Easy Hyb buffer at 42 °C overnight for hybridization. The membrane was detected using a ChemiDoc XRS+ system and Image Lab software.

### 4.10. HBx–Estradiol Benzoate Modeling

The crystallizing of HBx is futile. There is no detailed full-length structural information of HBx because of the formation of homodimers via disulfide bonds and acetylation [40]. HBx lacks a full-length sequence homology to any existing protein. Hence, the modeling of HBx was constructed by I-TASSER [41,42,43]. Then, the docking with estradiol benzoate was conducted using AutoDock 4 and AutoDockTools with the predicted model 1 of HBx [44]. After deleting water and adding hydrogens to the HBx, hydrogens were added to estradiol benzoate to modify it for docking. The docking was conducted, and the results were visualized using Pymol.

### 4.11. RNA-Seq

HepG2.2.15 cells were cultured in a 10 cm dish with 55 uM of estradiol benzoate or 0.55% DMSO DMEM. The RNA-seq libraries were sequenced using Illumina NovaSeq 6000 and PE150 (Novogene, Beijing, China). The filtered reads were mapped to the human reference genome hg38. The FPKM of each gene was obtained by HTSeq v0.9.1. The differentially expressed genes were analyzed by the edgeR package and were estimated according to the absolute fold change ≥ 1.5 and *p* < 0.05.

### 4.12. Statistical Analysis

Differences between the groups were examined using a T-test by GraphPad Prism. A two-sides *p*-value of <0.05 is considered statistically significant. Data are presented as means ± SD.

## 5. Conclusions

The SBHX21 cell line could be utilized for quick screening of the inhibitors of HBx, making it economically feasible to conduct large-scale screening of chemical libraries and identify new classes of inhibitors of HBx that restrain HBV cccDNA activity. Using the SBHX21 cell line screening system, we found that estradiol benzoate has functions in suppressing HBx and inhibiting HBV transcription and replication. Estradiol benzoate could be a novel antivirus molecular against HBV. The protein and molecular docking results revealed that estradiol benzoate interacts with HBx around the H-box domain of HBV. Fully understanding the interaction of estradiol benzoate and HBx requires further investigation.

## Figures and Tables

**Figure 1 molecules-27-05000-f001:**
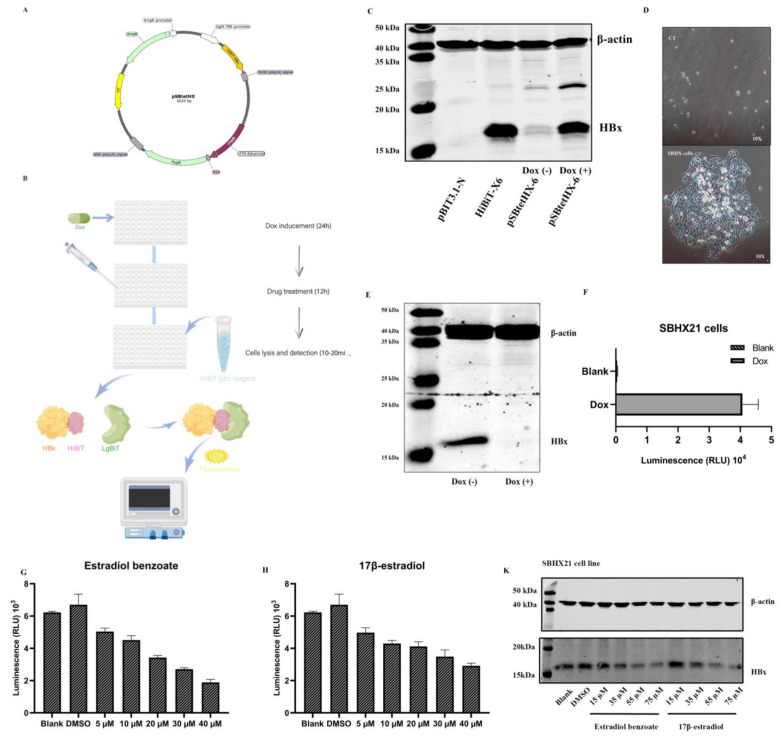
(**A**) Map of pSBtetHx plasmid. TRE: Tet-responsive promotor, HyGr: hygromycin. (**B**) Drug screening diagram drawn by Figdraw (www.figdraw.com, accessed on 30 April 2022). (**C**) Verification of HiBiT-X6 plasmid and pSBtetHX-6 plasmid in Huh7 cells by Western blot two days after transfection. The pSBtetHX-6 plasmid was divided into a Dox (1 µg/mL) treatment group and the blank group. (**D**) Upper panel (scale 10×): cells without plasmid transfection, lower panel (scale 10×): single-cell clone. (**E**) Expression of HBx from SBHX21 cell line with or without Dox treatment. (**F**) Luminescence results of SBHX21 with or without Dox treatment. (**G**,**H**) Luminescence results showed that estradiol benzoate and 17β-estradiol inhibit HBx levels. (**I**,**J**) The cell survival rate curve of estradiol benzoate and 17β-estradiol. Half-maximal inhibitory concentration (IC50) of estradiol benzoate was 279.0 µM, and IC50 of 17β-estradiol was 471.5 µM. (**K**) Western blot results of estradiol benzoate and 17β-estradiol showing inhibition of HBx in different concentrations. (**L**) Three independent replications of Western blot results were analyzed using ImageJ. Relative repressions of HBx were calculated by the ratio of HBx to β-actin. DMSO: dimethyl sulfoxide, Dox: doxycycline, NS: no significant, * *p* < 0.05, ** *p* < 0.01.

**Figure 2 molecules-27-05000-f002:**
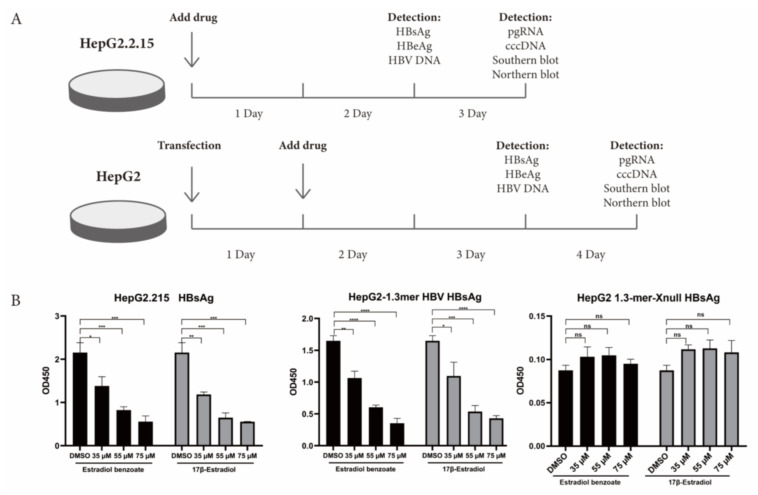
Estradiol benzoate and 17β-estradiol inhibit HBV transcription and replication. (**A**) Schematic diagram of cell processing and detection. (**B**) ELISA assay of HBsAg levels of estradiol benzoate– and 17β-estradiol–treated HepG2.2.15 cells, 1.3mer HBV–transfected HepG2 cells and 1.3mer HBV X-null plasmid transfected HepG2 cells in different concentrations. (**C**) ELISA assay of HBeAg levels of estradiol benzoate– and 17β-estradiol–treated HepG2.2.15 cells, 1.3mer HBV–transfected HepG2 cells, and 1.3mer HBV X-null plasmid transfected HepG2 cells in different concentrations. (**D**) Quantitative assay of supernatant HBV DNA. DMSO: dimethyl sulfoxide, NS: no significant, * *p* < 0.05, ** *p* < 0.01, *** *p* < 0.001, **** *p* < 0.0001.

**Figure 3 molecules-27-05000-f003:**
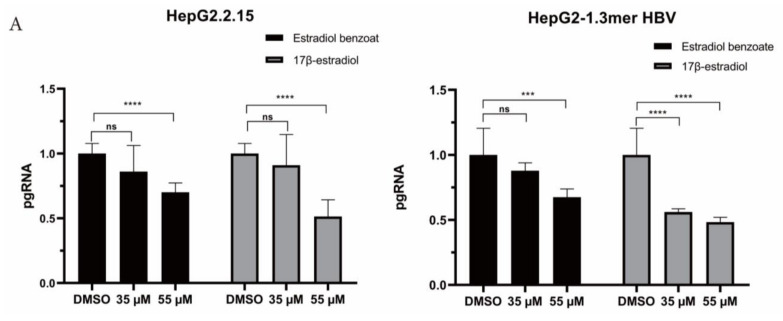
Estradiol benzoate inhibits HBV replication and transcription. (**A**) RT-qPCR of pgRNA level of estradiol benzoate– and 17β-estradiol–treated HepG2.2.15 cells and 1.3mer HBV–transfected HepG2 cells. (**B**) qPCR results of intracellular cccDNA of estradiol benzoate–treated HepG2.2.15 cells and 1.3mer HBV–transfected HepG2 cells. (**C**) Relative pgRNA level of estradiol benzoate–treated 1.3mer HBV X-null transfected HepG2 cells. (**D**) Northern blot results of estradiol benzoate–treated HepG2.2.15 cells and estradiol benzoate–treated 1.3mer HBV plasmid–transfected HepG2 cells. (**E**) Southern blot results of estradiol benzoate–treated 1.3mer HBV plasmid–transfected HepG2 cells. DMSO: dimethyl sulfoxide, kb: kilobase, RC: relaxed circular, DL: double strand, SS: single strand, NS: no significant, * *p* < 0.05, *** *p* < 0.001, **** *p* < 0.0001.

**Figure 4 molecules-27-05000-f004:**
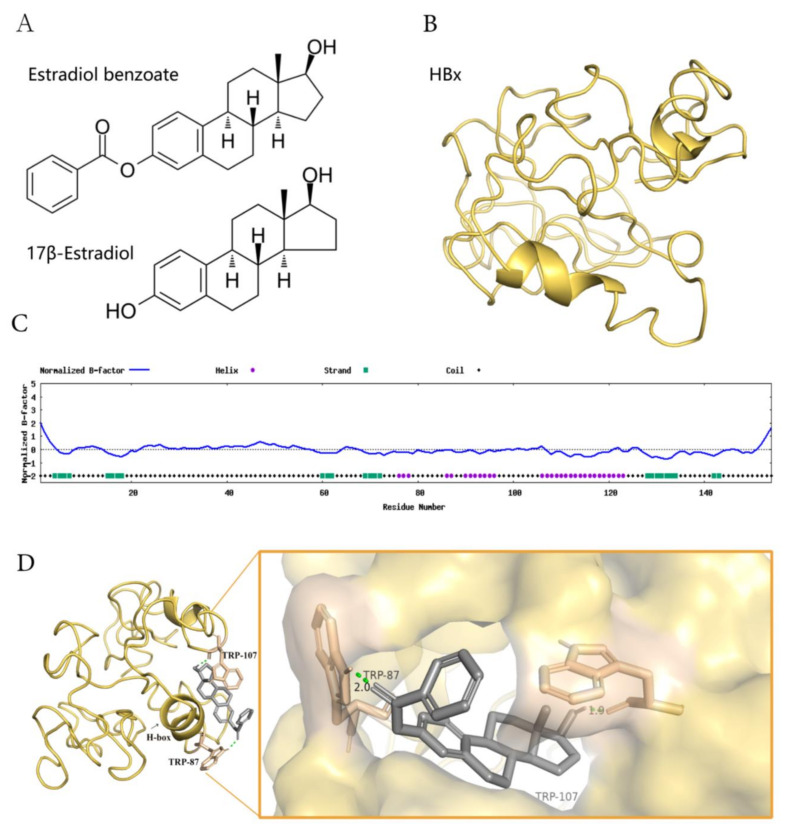
The model of estradiol benzoate and HBx interaction. (**A**) Chemical structures of estradiol benzoate and 17β-estradiol. (**B**) Predicted model 1 of HBx protein using I-TASSER. (**C**) Predicted normalized B-factor of HBx illustrating the helix and strands of HBx. (**D**) The estradiol benzoate docking model with HBx. Estradiol benzoate binds with HBx at Tryptophan 87 and Tryptophan 107. (**E**) A schematic model of HBx and DDB-1 interaction. DDB-1 uses its BPB domain to bind with CLU4. HBx binds to DDB-1 BPA-BPC double propeller with its H-box motif. (**F**) RNA-seq counts to DDB-CUL4 related downstream genes after being treated with estradiol benzoate.

## Data Availability

Not applicable.

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
