# Peer review of "Identification of Estradiol Benzoate as an Inhibitor of HBx Using Inducible Stably Transfected HepG2 Cells Expressing HiBiT Tagged HBx"

_molecules, 2022, doi:10.3390/molecules27155000_

Round 1
Reviewer 1 Report
Authors aimed to evaluate whether estradiol benzoate inhibits HBx protein and HBV transcription and replication.
There are several comments to be addressed
1) The possible hypothesis should be discussed more in-depth.
2) What about its role of HBV entry into the hepatocyte?
3) For estradiol benzoate to be available in the routine practice to treat HBV, the safety should be also a concern.
So, its potential harmful effect and how to deliver estradiol benzoate into hepatocyte should be further studeid. These issues should be discussed.
Author Response
Thank you for your time and helpful advice on this manuscript. we really appreciate that. If you have other comments that want to give us, please do not hesitate to contact us. Response please see the attachment.

Reviewer 2 Report
In this study, He et al. established the platform for HBx-specific antiviral screening, and they found that estradiol benzoate is one of the candidates. They tested the effect of estradiol benzoate in HepG2.2.15 and 1.3-mer transfection systems. However, some issues need to be addressed as the conclusions are not well-supported by the data. In addition, some important experiments are missing.
(1) In Fig.2, the decrease of HBsAg and HBeAg but no decrease in HBV DNA suggests that estradiol benzoate and 17-B-estradiol have no effects on DNA replication. How do the authors conclude the drugs affect DNA replication?
(2) In Fig.3, how the cccDNA quantification was done? What does the IU/ml mean for cccDNA? It is problematic.
(3) The authors should include the HBx-minus 1.3 mer to confirm their prediction that estradiol benzoate and 17-B-estradiol target to HBx by docking modeling. Without this data, the conclusion is not convincing.
(4) The authors should also include the experiment of HBV (WT and HBx-minus virus) infection in HepG2-NTCP cells and treat the infected cells with two compounds.
Author Response
Thank you for your time and helpful advice on this manuscript. we really appreciate that. May we apologize for not being able to finished all the experiments required due to the time limit. If you still want us to conduct those experiments, we are more than willing to communicate with you and the editor to request more time to perform those experiments. If you have other comments that want to give us, please do not hesitate to contact us. Response please see the attachment.

Round 2
Reviewer 1 Report
Authors addressed raiaed issues appropriately.
Reviewer 2 Report
The revised article looks fine.